# AMMD: Attentive Maximum Mean Discrepancy for Few-Shot Image Classification

## Abstract

Metric-based methods have attained promising performance for the few-shot classification of images. Maximum Mean Discrepancy (MMD) is a typical distance between distributions, requiring to compute expectations w.r.t. data distributions. In this paper, we propose Attentive Maximum Mean Discrepancy (AMMD) to assist MMD with distributions adaptively estimated by an attentive distribution generation module. Based on AMMD, the few-shot learning is modeled as the AMMD metric learning problem. In implementation, we incorporate the part-based feature representation for modeling the AMMD between images. By meta-learning technique, the attentive distribution generation module of AMMD can be learned to generate feature distributions for computing MMD between images, with higher probability mass on the more discriminative features. In the meta-test phase, each query image is labeled as the support class with minimal AMMD to the query image. Extensive experiments show that our AMMD achieves competitive or state-of-the-art performance on few-shot classification benchmark datasets of *mini*ImageNet, *tiered*ImageNet, CIFAR-FS, and FC100.

## 1 Introduction

Deep learning has made remarkable progress across a range of computer vision tasks, such as image classification (He et al., 2016), object detection (Redmon et al., 2016), and image semantic segmentation (Chen et al., 2017). However, deep learning methods typically rely on a large amount of labeled data for training neural networks. In contrast, humans can quickly learn a new concept from a limited amount of labeled samples. To bridge the gap between humans and deep learning methods in scenarios with limited labeled data, few-shot learning (Fei-Fei et al., 2006; Koch et al., 2015; Wertheimer et al., 2021; Guo et al., 2022) has gained significant attention in recent years.

As one of the mainstream methods for few-shot learning, metric-based methods first embed images into a feature space, then measure distances between query images and support images in the feature space, and finally predict labels of query images based on their distances to support images. One of the main concerns of metric-based methods is how to appropriately measure the distance of images in feature space. In the earlier works (Vinyals et al., 2016; Snell et al., 2017; Hu et al., 2018), the distance is measured among global representations of images in the feature space. Due to the images being scarce in few-shot learning, the global representations might not be sufficient to represent the images, which further limits the performance of these methods. Recently, some methods have been proposed to measure the distance defined on distributions of local features of images (Li et al., 2021; Xie et al., 2022; Zhang et al., 2023). Compared to earlier works based on global representations, these recent methods achieve better performance, since the local features are more diverse and contain richer discriminative information.

Maximum Mean Discrepancy (MMD) (Gretton et al., 2012), as a typical distance defined on distributions, can be naturally applied to few-shot classification. Intuitively, we can take advantage of MMD to measure the distance between the distribution of a query image and the distribution of a class of support images in the feature space. However, the distributions of query images and support images are practically unknown, which makes it intractable to compute the expectations in MMD. A popular alternative to MMD is empirical MMD, which empirically approximates the expectation with the mean of features. However, empirical MMD might not be an optimal choice for few-shot classification because the features of an image are not uniformly important for recognition.

Different features of an image, containing different discriminative information, should contribute differently to the distance measure. Therefore, empirical MMD might hinder the performance of few-shot classification.

In this paper, we propose Attentive Maximum Mean Discrepancy (AMMD) for few-shot classification. Instead of using the mean of patch-level features to approximate expectations as in empirical MMD, AMMD adaptively estimates distributions of patch-level features by an attentive distribution generation module (ADGM) and then computes the expectations in MMD based on the estimated distributions. The ADGM can learn to adaptively produce feature distributions under the framework of meta-learning, and the learned distributions put more probability mass on the more discriminative features. As a result, the discriminative features contribute more to the distance of AMMD, which makes AMMD potentially achieve better performance for few-shot classification than empirical MMD. To generate the distribution by the ADGM, the patch-level features are sequentially processed by self-attention block, global average pooling, and cross-attention block.

To verify the efficacy of the proposed AMMD for few-shot classification, we conduct comprehensive experiments on various datasets including *mini*ImageNet (Vinyals et al., 2016), *tiered*ImageNet (Ren et al., 2018), CIFAR-FS (Bertinetto et al., 2019), and FC100 (Oreshkin et al., 2018). In these experiments, we employ ResNet-12 (He et al., 2016) and Swin-Transformer (Liu et al., 2021b) as the backbone networks. The experimental results demonstrate that the proposed AMMD can match or outperform the current state-of-the-art performance in few-shot classification.

## 2 RELATED WORK

### 2.1 FEW-SHOT LEARNING METHODS

**Metric-based methods** focus on learning to measure distances between images in the latent feature space and perform classification based on the distances. ProtoNet (Snell et al., 2017), MatchingNet (Vinyals et al., 2016), and RelationNet (Hu et al., 2018) measure distances w.r.t. global representations of images in the feature space by respectively employing Euclidean distances, cosine similarity, and learnable network as metrics. Different to them, some methods are proposed to measure distances w.r.t. distributions defined on all patch-level features of images, such as ADM (Li et al., 2021), DeepBDC (Xie et al., 2022) and DeepEMD (Zhang et al., 2023). ADM (Li et al., 2021) assumes that patch-level features obey multivariate Gaussian distribution, and takes KL divergence as the metric. DeepEMD (Zhang et al., 2023) utilizes Earth Mover's Distance to measure distances. Though taking the discrimination of patch-level features into consideration, DeepEMD is computationally expensive, due to the iterative algorithm for linear programming. DeepBDC (Xie et al., 2022) proposes to utilize Brownian Distance Covariance which is based on the Fourier transformation of the joint of probability density functions of images.

**Optimization-based methods** aim to develop optimization strategies that enable the neural networks to quickly adapt to unseen few-shot learning tasks. MAML (Finn et al., 2017) is designed to learn good initialization of neural networks that can adapt to unseen tasks by one-step gradient descent. Inspired by MAML, numerous follow-up works were proposed to find better initialization of neural networks (Rusu et al., 2018; Lee et al., 2019; Jamal & Qi, 2019; Oh et al., 2021).

**Generation-based methods** aim to enlarge the training set through data generation and argumentation. Zhang et al. (2018) and Yang et al. (2022) propose to generate images as training data by Generative Adversarial Networks (Goodfellow et al., 2014). Though the generated images might not be realistic, they are useful to improve the performance of few-shot classification. Yang et al. (2021) and Guo et al. (2022) produce extra features from calibrated distributions of novel classes, where the calibrated distributions are transferred from distributions of base classes.

### 2.2 MAXIMUM MEAN DISCREPANCY

Maximum Mean Discrepancy (MMD) was originally proposed in (Gretton et al., 2006) as a kernel-based test statistic for the two-sample problem. Recent works have extended MMD into several fields, such as few-shot learning (Liu et al., 2021a; Chowdhury & Bathula, 2022), generative models (Li et al., 2015; 2017; Wang et al., 2019), and unsupervised domain adaptation (UDA) (Long et al., 2015; 2017; Yan et al., 2017; Ren et al., 2020).

In few-shot learning, PDA (Liu et al., 2021a) uses domain-level MMD loss and class-level MMD loss to optimize the neural network. IPNet (Chowdhury & Bathula, 2022) relies on MMD in medical imaging to reweight support images for generating support class prototypes. These methods apply the empirical MMD to support images from the perspective of global features. MMD was also applied to generative models to measure the distributional distance between the generated images and real images (Li et al., 2015). The methods in Li et al. (2017) and Wang et al. (2019) further extend MMD as a loss function in the GAN models. In UDA task, MMD is typically used as a loss term to reduce the discrepancy between the source and target domains (Long et al., 2015; 2017; Ren et al., 2020). Yan et al. (2017) proposed a weighted MMD to generate class-specific auxiliary weights for images in source domain. The auxiliary weights are similar to our generated feature distributions, however, we generate the distributions adaptively from the support and query images using attentive distribution generation module, for few-shot classification.

Compared with these works based on MMD, we focus on the few-shot image classification task. Our AMMD builds on MMD by using adaptively generated feature distributions to compute the expectations of features to measure the distance between different images. In addition, we extract patch-level features from images that contain rich local information rather than global features.

## 3 METHOD

In the context of few-shot classification, the network should be able to recognize unseen query images based on a few labeled support images. To this end, the existing few-shot learning methods are typically formulated within the meta-learning (Finn et al., 2017) framework by learning in an episodic manner. In each episodic, there is a $N$-way $K$-shot task, where the task consists of a support set $\mathcal{S}$ and a query set $\mathcal{Q}$. In the support set $\mathcal{S}$, there are $N$ classes and $K$ labeled images per class. By denoting $\mathcal{S}^c = \{(X_i^s, Y_i^s = c)\}_{i=1}^K$ as the set of images from the $c$-th class with $X_i^s$ as a support image and $Y_i^s$ as the corresponding label, the support set can be represented as $\mathcal{S} = \{\mathcal{S}^1, \ldots, \mathcal{S}^N\}$. In the meta-training phase, the labels of images in the query set $\mathcal{Q}$ are known and each class consists of $K_Q$ images. We here represent the query set $\mathcal{Q} = \{X_i^q, Y_i^q\}_{i=1}^{NK_Q}$ with $X_i^q$ as a query image and $Y_i^q$ as the corresponding label. For brevity, we may omit the subscript in the query image $X_i^q$ and the corresponding label $Y_i^q$.

We propose a new metric, dubbed Attentive Maximum Mean Discrepancy (AMMD), for few-shot classification. The proposed AMMD is used to measure the distance between distributions of support images and query images in the feature space. AMMD assists MMD with an attentive distribution generation module (ADGM) to adaptively estimate the unknown distribution of images in the feature space. In Fig. 1, we illustrate the AMMD for few-shot classification. We first extract the set of patch-level features from the query image and the support image by a patch-level feature extractor. For a query image $X^q$, we split it into multiple patches, and obtain features of each patch by feeding all patches into the neural network $g_\theta$. All patch-level features of the query image $X^q$ constitute the query feature set $F^q$. Likewise, we can obtain multiple patch-level features of each support image via the shared neural network $g_\theta$, and all patch-level features of all images belong to the $c$-th class constitutes the support feature set $F^{s,c}$ of the $c$-th class. After obtaining the query feature set $F^q$ and the support feature set of each class (e.g., $F^{s,c}$), we compute the distance between the distribution of $F^q$ and the distribution of $F^{s,c}$ by AMMD, in which the ADGM in AMMD adaptively estimates the distributions. Based on the distance, we design a triplet loss for meta-training and categorize the query image as the nearest class in the meta-test phase.

In this section, we first describe how to extract patch-level features of images by patch-level feature extractor in Sec. 3.1, then introduce details of the proposed AMMD in Sec. 3.2, and finally present the meta-training loss based on AMMD for few-shot learning in Sec. 3.3.

### 3.1 PATCH-LEVEL FEATURE EXTRACTOR

The proposed AMMD relies on the distributions of images in feature space, thus the features of images are supposed to be diverse and representative. As shown in the left of Fig. 1, we evenly split each image into multiple patches, then obtain patch-level features by feeding these patches into the neural network $g_\theta$. By this means, we obtain the query feature set $F^q = \{f_i^q\}_{i=1}^M$ of the query image $X^q$ with $f_i^q \in \mathbb{R}^d$, where $M$ is the number of patch-level features and $d$ is the feature

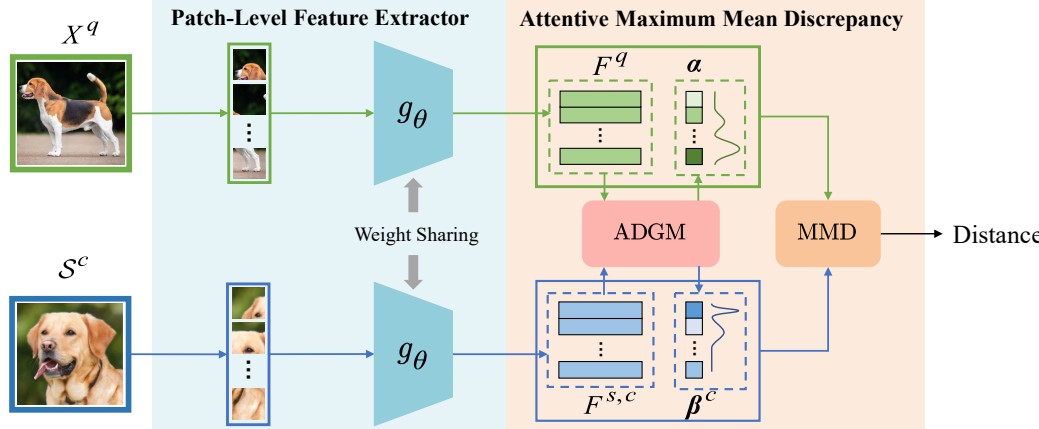

Figure 1: An overview of method. A support class $\mathcal{S}^c$ (1-shot) and a query image $X^q$ are split into patches and then fed into the backbone to extract their patch-level local features $F^{s,c}$ and $F^q$. After that, the Attention-based Distribution Generation Module (ADGM) generates the corresponding local feature distributions $\boldsymbol{\alpha}$ for $F^q$ and $\boldsymbol{\beta^c}$ for $F^{s,c}$. Finally, the Maximum Mean Discrepancy (MMD) combines $F^q$ with $\boldsymbol{\alpha}$ and $F^{s,c}$ with $\boldsymbol{\beta^c}$ to perform the calculation to measure the distance between the query image $X^q$ and the support class $\mathcal{S}^c$.

dimension. Meanwhile, we can obtain the support feature set of each class by gathering patch-level features of all images belonging to the class. For example, the support feature set of the $c$-th class is $F^{s,c} = \{f_j^{s,c}\}_{j=1}^{KM}$ with $f_j^{s,c} \in \mathbb{R}^d$.

## 3.2 AMMD FOR FEW-SHOT LEARNING

After obtaining the query feature set $F^q$ and the support feature set $F^{s,c}$ ($c = 1, \ldots, N$), we use the proposed Attentive Maximum Mean Discrepancy (AMMD) to measure the distance between the distribution $P^q$ of the query feature set $F^q$ and the distribution $P^{s,c}$ of the support feature set $F^{s,c}$ ($c = 1, \ldots, N$). In this subsection, we will discuss the limitations of directly applying MMD to few-shot classification, following which we introduce the proposed AMMD thoroughly.

### 3.2.1 LIMITATIONS OF MMD FOR FEW-SHOT CLASSIFICATION

Though we can directly apply MMD to few-shot classification, there are still some limitations remaining to overcome. Specifically, MMD can measure the distance between the distribution $P^q$ of the query feature set $F^q$ and the distribution $P^{s,c}$ of the support feature set $F^{s,c}$ ($c = 1, \ldots, N$) as

$$\mathrm{MMD}\left(P^q, P^{s,c}\right) = \sup_{\|h\|_{\mathcal{H}} \leq 1} \left(\mathbb{E}_{f^q \sim P^q}[h\left(f^q\right)] - \mathbb{E}_{f^{s,c} \sim P^{s,c}}[h\left(f^{s,c}\right)]\right), \tag{1}$$

where $h$ is a function in a reproducing kernel Hilbert space (RKHS) $\mathcal{H}$, and $\mathbb{E}$ is the expectation operator. Then the square of the MMD distance can be computed as

$$\mathrm{MMD}^2\left(P^q, P^{s,c}\right) = \|\mathbb{E}_{f^q \sim P^q}[\phi(f^q)] - \mathbb{E}_{f^{s,c} \sim P^{s,c}}[\phi(f^{s,c})]\|_{\mathcal{H}}^2, \tag{2}$$

where $\phi$ is the canonical feature map in RKHS. It is worth mentioning that $\phi$ is not a neural network in this paper but defines the kernel function by $k(x, y) = \langle \phi(x), \phi(y) \rangle$. In practice, the distributions $P^q$ and $P^{s,c}$ are unknown, thus it is intractable to compute the square of MMD in Eq. (2). The most popular method of estimating MMD is empirical MMD utilizing empirical expectations to approximate population expectations

$$\mathrm{MMD}^2\left(X^q, \mathcal{S}^c\right) = \|\frac{1}{M} \sum_{i=1}^{M} \phi(f_i^q) - \frac{1}{KM} \sum_{j=1}^{KM} \phi(f_j^{s,c})\|_{\mathcal{H}}^2. \tag{3}$$

However, empirical MMD treats each patch-level feature with equal importance, ignoring the fact that different patches contain different discriminative information for few-shot classification.

### 3.2.2 AMMD

To overcome the limitations of empirical MMD for few-shot classification, we develop a new variant of MMD, dubbed AMMD, as shown in the right of Fig. 1. Different from empirical MMD, we adaptively estimate the distribution $P^q$ of the query feature set $F^q$ and the distribution $P^{s,c}$ of the support feature set $F^{s,c}$ ($c = 1, \ldots, N$). Due to that $P^q$ and $P^{s,c}$ are discrete distributions, we can respectively represent them as

$$P^q(f^q) = \sum_{i=1}^{M} \alpha_i \delta_{f_i^q}(f^q) \quad \text{and} \quad P^{s,c}(f^{s,c}) = \sum_{i=1}^{KM} \beta_i^c \delta_{f_j^{s,c}}(f^{s,c}), \tag{4}$$

where $\delta_{f_i^q}$ and $\delta_{f_j^{s,c}}$ are Dirac function, $\sum_{i=1}^{M} \alpha_i = 1$, $\sum_{j=1}^{KM} \beta_j^c = 1$. By substituting the discrete distributions $P^q$ and $P^{s,c}$ into Eq. (2), we can easily obtain

$$\text{AMMD}^2\left(P^q, P^{s,c}\right) = \left\| \sum_{i=1}^{M} \alpha_i \phi\left(x_i^q\right) - \sum_{j=1}^{KM} \beta_j^c \phi\left(x_{j,c}^s\right) \right\|_{\mathcal{H}}^2. \tag{5}$$

Considering that $k(x, y) = \langle \phi(x), \phi(y) \rangle$, we further have

$$\text{AMMD}^2\left(P^q, P^{s,c}\right) = \sum_{i=1}^{M}\sum_{j=1}^{M} \alpha_i \alpha_j k\left(f_i^q, f_j^q\right) + \sum_{i=1}^{KM}\sum_{j=1}^{KM} \beta_i^c \beta_j^c k\left(f_i^{s,c}, f_j^{s,c}\right)$$
$$- 2\sum_{i=1}^{M}\sum_{j=1}^{KM} \alpha_i \beta_j^c k\left(f_i^q, f_j^{s,c}\right). \tag{6}$$

By denoting $\boldsymbol{\alpha} = [\alpha_1, \ldots, \alpha_M]$, $\boldsymbol{\beta}^c = [\beta_1^c, \ldots, \beta_{KM}^c]$, we can represent Eq. (6) in a compact form:

$$\text{AMMD}^2\left(P^q, P^{s,c}\right) = \boldsymbol{\alpha}^\top K^{qq} \boldsymbol{\alpha} + \boldsymbol{\beta}^{c\top} K^{ss,c} \boldsymbol{\beta}^c - 2\boldsymbol{\alpha}^\top K^{qs,c} \boldsymbol{\beta}^c, \tag{7}$$

where $K_{qq}$ is the kernel matrix on the query feature set with $[K^{qq}]_{ij} = k(f_i^q, f_j^q)$, $K^{ss,c}$ is the kernel matrix on the support feature set with $[K^{ss,c}]_{ij} = k(f_i^{s,c}, f_j^{s,c})$, and $K^{qs,c}$ is the kernel matrix with $[K^{qs,c}]_{ij} = k(f_i^q, f_j^{s,c})$. Moreover, we mainly use the linear kernel $k(x, y) = x^\top y$ in our AMMD.

To adaptively estimate the distribution $P^q$ of the query feature set $F^q$ and the distribution $P^{s,c}$ of the support feature set $F^{s,c}$ ($c = 1, \ldots, N$), i.e. $\boldsymbol{\alpha}$ and $\boldsymbol{\beta}^c$, we design an attentive distribution generation module (ADGM), which will be introduced in Sec. 3.2.3.

### 3.2.3 ATTENTIVE DISTRIBUTION GENERATION MODULE IN AMMD

The attentive distribution generation module (ADGM) is designed for adaptively estimating the parameters $\boldsymbol{\alpha}$ and $\boldsymbol{\beta}^c$ of the distributions $P^q$ and $P^{s,c}$. The parameters $\boldsymbol{\alpha}$ and $\boldsymbol{\beta}^c$ are further utilized by AMMD in Eq. (7). As shown in Fig. 2, ADGM consists of operations of a self-attention block, global average pooling, and a cross-attention block. After individually feeding the query feature set $F^q$ and the support feature set $F^{s,c}$ into the self-attention block, we utilize global average pooling to respectively produce the global query feature and global support feature. The global query feature refers to the global representation of the query image, and the global support feature refers to the global representation of all support images

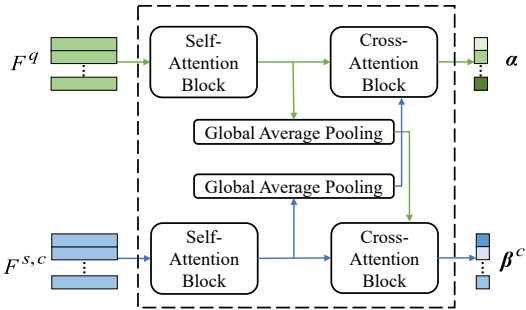

Figure 2: Illustration of the Attention-based Distribution Generation Module (ADGM).

belonging to the same class (e.g., the $c$-th class). Subsequently, we use the cross-attention block to generate the distribution $P^q$ and $P^{s,c}$ parameterized by $\boldsymbol{\alpha}$ and $\boldsymbol{\beta}^c$ of the query feature set $F^q$ and support feature set $F^{s,c}$ respectively.

**Self-attention block.** In the self-attention block, the query feature set $F^q$ is processed by

$$\hat{F}^q = \frac{F^q + \text{MHS}(F^q)}{\|F^q + \text{MHS}(F^q)\|_2},\tag{8}$$

where "MHS" represents the multi-head self-attention operation (Vaswani et al., 2017). By replacing the input $F^q$ with the support feature set $F^{s,c}$, we can obtain the processed feature $\hat{F}^{s,c}$ of the support feature set $F^{s,c}$.

**Global average pooling.** Based on the processed features $\hat{F}^q$ and $\hat{F}^{s,c}$, we utilize global average pooling to get global representation $G^q$ (resp. $G^{s,c}$) of the query image (resp. the support images from $c$-th class).

**Cross-attention block.** With the processed features $\hat{F}^q$ and the global support representation $G^{s,c}$, we employ the cross-attention block to estimate $\boldsymbol{\alpha}$ in the distribution of the query feature set $P^q$ by

$$\boldsymbol{\alpha} = \text{softmax}\left(\frac{(\hat{F}^q W^q)(G^{s,c} W^s)^\top}{\tau}\right),\tag{9}$$

where $W^q$ and $W^s$ are learnable parameters, $\tau$ is the temperature. Similarly, we can produce $\boldsymbol{\beta^c}$ in the distribution of the support feature set $P^{s,c}$ by

$$\boldsymbol{\beta^c} = \text{softmax}\left(\frac{(\hat{F}^{s,c} W^s)(G^q W^q)^\top}{\tau}\right).\tag{10}$$

By such ways, we put more probability mass on the patch-level feature in the query feature set (resp. support feature set) if the patch-level feature is more similar to the global support (resp. query) representation. As a result, the AMMD puts more emphasis on the discriminative patch-level features when computing the MMD between distributions.

### 3.3 META-TRAINING LOSS

In the meta-training phase, we take advantage of triplet loss to train the patch-level feature extractor and the attentive distribution generation module. The triplet loss is defined as

$$L = \frac{1}{NK_Q}\sum_{i=1}^{NK_Q} \max(0, \gamma + \text{AMMD}^2(P_i^q, P^{s,+}) - \text{AMMD}^2(P_i^q, P^{s,-})),\tag{11}$$

where $\gamma$ is a margin, $P_i^q$ is the distribution of the query feature set $F_i^q$ obtained by feeding the query image $X_i^q$ into the patch-level feature extractor, $P^{s,+}$ is the distribution of the support feature set corresponding to images from the same class of the query image $X_i^q$, and $P^{s,-}$ is the distribution of a support feature set corresponding to images from a negative class. In this paper, the negative class is taken as the nearest support class with different class label to the query image $X_i^q$, measured in AMMD distance.

## 4 EXPERIMENTS

### 4.1 IMPLEMENTATION DETAILS

**Datasets.** We evaluate the effectiveness of our method on four popular few-shot classification datasets: *mini*ImageNet (Vinyals et al., 2016), *tiered*ImageNet (Ren et al., 2018), CIFAR-FS (Bertinetto et al., 2019), and FC100 (Oreshkin et al., 2018). *mini*ImageNet and *tiered*ImageNet are two subsets from ImageNet (Deng et al., 2009). CIFAR-FS and FC100 are derived from CIFAR100 (Krizhevsky et al., 2009). We provide further dataset details in Appendix A.1.

**Backbone.** For a fair comparison, we employ ResNet-12 (He et al., 2016) as one of the backbone networks, which is widely used in previous works (Liu et al., 2022; Xie et al., 2022; Zhang et al., 2023). Furthermore, to confirm the effectiveness of our method in transformer-based networks, we use Swin-Tiny (Liu et al., 2021b) as an additional backbone network. For few-shot classification, the results of using transformer-based backbones in FewTRUE (Hiller et al., 2022) show that Swin-Tiny can achieve higher performance than ViT-S (Dosovitskiy et al., 2020).

Table 1: Few-shot classification accuracy (%) on *mini*ImageNet and *tiered*ImageNet.

| Method | Backbone | *mini*ImageNet | | *tiered*ImageNet | |
|---|---|---|---|---|---|
| | | 5-way 1-shot | 5-way 5-shot | 5-way 1-shot | 5-way 5-shot |
| ProtoNet (Snell et al., 2017) | *ResNet-12* | 60.76±0.47 | 78.51±0.34 | 68.29±0.52 | 83.59±0.37 |
| MetaOptNet (Lee et al., 2019) | *ResNet-12* | 62.64±0.82 | 78.63±0.46 | 65.99±0.72 | 81.56±0.53 |
| FEAT (Ye et al., 2020) | *ResNet-12* | 66.78±0.20 | 82.05±0.14 | 70.80±0.23 | 84.79±0.16 |
| FRN (Wertheimer et al., 2021) | *ResNet-12* | 66.45±0.19 | 82.83±0.13 | 71.16±0.22 | 86.01±0.15 |
| PDA (Liu et al., 2021a) | *ResNet-12* | 65.75±0.43 | 83.37±0.30 | 72.28±0.49 | 86.70±0.33 |
| Meta-DeepBDC (Xie et al., 2022) | *ResNet-12* | 67.34±0.43 | 84.46±0.28 | 72.34±0.49 | 87.31±0.32 |
| MCL (Guo et al., 2022) | *ResNet-12* | 69.31±0.21 | 85.11±0.18 | 73.62±0.24 | 86.29±0.19 |
| DeepEMD (Zhang et al., 2023) | *ResNet-12* | 67.83±0.29 | 83.14±0.57 | 73.13±0.32 | 87.08±0.60 |
| STANet (Dong et al., 2023b) | *ResNet-12* | 69.84±0.47 | 84.88±0.30 | 73.08±0.49 | 86.80±0.34 |
| RENet-ventral (Dong et al., 2023a) | *ResNet-12* | 69.71±0.45 | 84.23±0.29 | 73.94±0.48 | 87.15±0.35 |
| AMMD(ours) | *ResNet-12* | **70.31±0.45** | **85.22±0.29** | **74.22±0.50** | **87.55±0.34** |
| FewTrue (Hiller et al., 2022) | *Swin-Tiny* | 70.52±0.84 | 84.84±0.56 | 76.49±0.92 | 89.36±0.58 |
| AMMD(ours) | *Swin-Tiny* | **71.31±0.45** | **86.07±0.29** | **77.35±0.48** | **89.49±0.31** |

Table 2: Few-shot classification accuracy (%) on CIFAR-FS and FC100.

| Method | Backbone | CIFAR-FS | | FC100 | |
|---|---|---|---|---|---|
| | | 5-way 1-shot | 5-way 5-shot | 5-way 1-shot | 5-way 5-shot |
| ProtoNet (Snell et al., 2017) | *ResNet-12* | 69.45±0.50 | 83.69±0.35 | 39.45±0.39 | 58.02±0.43 |
| MetaOptNet (Lee et al., 2019) | *ResNet-12* | 72.60±0.70 | 84.30±0.50 | 41.10±0.60 | 55.50±0.60 |
| ConstellationNet (Xu et al., 2021) | *ResNet-12* | 75.40±0.20 | 86.80±0.20 | 43.80±0.20 | 59.70±0.20 |
| Meta-NVG (Zhang et al., 2021) | *ResNet-12* | 74.63±0.91 | 86.90±0.50 | 44.60±0.70 | 60.90±0.60 |
| MixFSL (Zhang et al., 2021) | *ResNet-12* | - | - | 44.89±0.63 | 60.70±0.60 |
| RENet (Kang et al., 2021) | *ResNet-12* | 74.51±0.46 | 86.60±0.32 | | |
| DeepEMD (Zhang et al., 2023) | *ResNet-12* | 73.31±0.29 | 85.43±0.37 | **45.23±0.26** | **61.39±0.76** |
| RENet-ventral (Dong et al., 2023a) | *ResNet-12* | 75.82 | 87.45 | - | - |
| AMMD(ours) | *ResNet-12* | **75.92±0.48** | **87.95±0.32** | 44.95±0.40 | **61.39±0.43** |
| FewTURE (Hiller et al., 2022) | *Swin-Tiny* | 76.52±0.85 | 87.28±0.65 | 45.61±0.75 | 62.35±0.74 |
| AMMD(ours) | *Swin-Tiny* | **77.61±0.46** | **88.08±0.33** | **46.86±0.43** | **63.79±0.44** |

**Training.** The training phase contains pre-training and meta-training, as done in recent works (Liu et al., 2022; Zhang et al., 2023; Lin et al., 2023; Hiller et al., 2022). In the pretraining phase, ResNet-12 and Swin-Tiny are trained on the whole meta-training dataset by respectively employing supervised classification and the self-supervision method of iBOT (Zhou et al., 2022). In the meta-training phase, taking the pre-trained networks as initialization, we train ResNet-12 and Swin-Tiny with SGD. To produce patch-level features with ResNet-12, we crop each image into a multi-scale grid of size $3 \times 3 + 2 \times 2$ and resize the multi-scale patches in the grid cell as $84 \times 84$, then feed these resized patches into ResNet-12 to obtain $3 \times 3 + 2 \times 2$ patch-level features. For Swin-Tiny, we resize each image as $224 \times 224$, then feed the resized image into Swin-Tiny to get $7 \times 7$ patch-level features. More details of pre-training and meta-training can be found in Appendix A.2 and Appendix A.3 respectively. In addition, the configurations of hyperparameters can also be found in Appendix A.3. All experiments are conducted on Nvidia V100 GPUs except that the pre-training of Swin-Tiny is conducted on 8 Nvidia A100 GPUs.

**Evaluation.** We evaluate the performance of AMMD on 5-way 1-shot and 5-way 5-shot settings with the top-1 accuracy as the evaluation metric. In each experiment, we randomly generate 2000 tasks from the test set, in which each task contains 15 query images per class. We report the average accuracy over the 2000 tasks at the corresponding 95% confidence interval. In Appendix A.4, we conduct more experiments for the influence of temperature $\tau$ and the influence of kernel type.

## 4.2 COMPARISON WITH STATE-OF-THE-ART METHODS

In Table 1 and Table 2, we demonstrate the results of the proposed AMMD on four benchmark datasets, including *mini*ImageNet, *tiered*ImageNet, CIFAR-FS, and FC100, with ResNet-12 and Swin-Tiny as backbone network.

When taking ResNet-12 as the backbone, the proposed AMMD achieves state-of-the-art performance on *mini*ImageNet, *tiered*ImageNet, and CIFAR-FS under the settings of 5-way 1-shot classi-

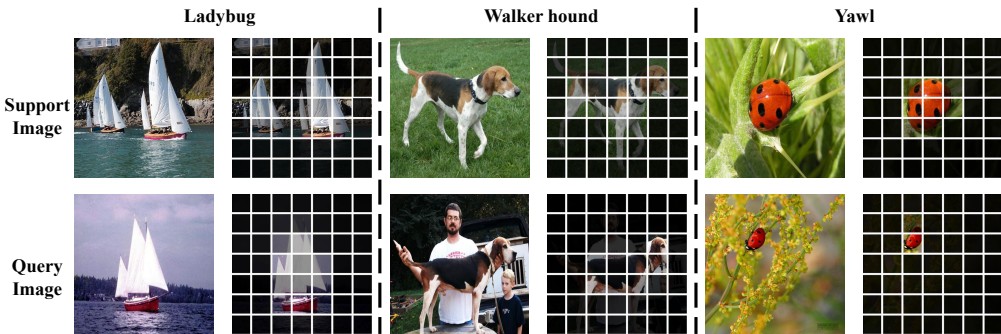

Figure 3: Visualizations of feature distributions generated by ADGM. We use Swin-Tiny to extract $7 \times 7$ patch-level features from each image to generate feature distributions. Given two images (upper and down), we show the probability mass corresponding to image patches, where brighter patches have higher probability mass. Our method can efficiently highlight the patch-level features that have discriminative information relative to the class of support/query image.

fication and 5-way 5-shot classification. On *mini*ImageNet, the accuracy value of AMMD is 0.47% (resp. 0.11%) higher than the accuracy value of STANet (resp. MCL) producing the second-best performance of 5-way 1-shot (resp. 5-way 5-shot) classification. On *tiered*ImageNet, RENet-ventral and Meta-DeepBDC produce the most competitive performance respectively of 5-way 1-shot classification and 5-way 5-shot classification, but their accuracy values are 0.28% and 0.24% lower than the accuracy values achieved by AMMD. On CIFAR-FS, our AMMD outperforms RENet-ventral 1-shot by 0.1% and 5-shot by 0.5%. On FC100, AMMD and DeepEMD obtain the best performance of 5-way 5-shot classification, while DeepEMD performs better than AMMD with a margin of 0.28% on 5-way 1-shot classification. However, AMMD outperforms DeepEMD on the other three benchmark datasets by a larger margin. Particularly, for 5-way 1-shot classification and 5-way-5shot classification on *mini*ImageNet, the accuracy values of AMMD are respectively 2.48% and 2.08% higher than the accuracy values of DeepEMD. Additionally, for 5-way 1-shot classification and 5-way-5shot classification on *tiered*ImageNet, AMMD achieves 2.61% and 2.52% higher accuracy values than DeepEMD.

When taking Swin-Tiny as the backbone, our AMMD achieves state-of-the-art performance on all datasets. To the best of our knowledge, only FewTRUE takes Swin-Tiny as the backbone network for few-shot classification, and its results are reported by running their codes. For the 5-way 1-shot classification, AMMD outperforms FewTURE by 0.79% on *mini*ImageNet, 0.86% on *tiered*ImageNet, 1.09% on CIFAR-FS, and 1.25% on FC100. For the 5-way 5-shot classification, our AMMD outperforms FewTURE by 1.23% on *mini*ImageNet, 0.13% on *tiered*ImageNet, 0.80% on CIFAR-FS, and 1.44% on FC100.

In summary, the proposed AMMD achieves the overall best performance among the competitors. Even though DeepEMD achieves competitive or even slightly better results than AMMD on FC100, the results of DeepEMD are inferior to AMMD on CIFAR-FS, *mini*ImageNet, and *tiered*ImageNet.

### 4.3 VISUALIZATION ANALYSIS

To show the effectiveness of the feature distributions adaptively estimated from ADGM, we associate the probability mass of distributions to the corresponding patches in images for visualization. As shown in Fig. 3, the distributions generated by ADGM generally assign larger probability mass to the patches containing discriminative image regions for classification. The proposed ADGM can adaptively find and highlight the patches in images that are semantically related to the target class.

### 4.4 ABLATION STUDY

**Influence of ADGM.** The influence of ADGM is shown in Table 3, and AMMD is equivalent to empirical MMD when AGDM is not used to generate distributions $\alpha$ and $\beta^c$. ADGM is used to adaptively estimate the distribution of features from support feature set and query feature set to help

Table 3: Ablation of ADGM by individually applying it on generating the query feature distribution $\alpha$ for query feature set $F^q$ and generating the support feature distribution $\beta^c$ for support feature set $F^{s,c}$. The baseline of AMMD without distributions $\alpha$ and $\beta^c$ is equivalent to empirical MMD.

| Backbone | AWGM | | *mini*ImageNet | | *tiered*ImageNet | |
|---|---|---|---|---|---|---|
| | $\alpha$ | $\beta^c$ | 5-way 1-shot | 5-way 5-shot | 5-way 1-shot | 5-way 5-shot |
| ResNet-12 | | | 69.34±0.46 | 82.49±0.34 | 72.49±0.49 | 83.68±0.40 |
| | ✓ | | 69.41±0.46 | 85.16±0.30 | 73.26±0.52 | 87.46±0.34 |
| | | ✓ | 69.90±0.45 | 84.93±0.30 | 73.24±0.50 | 87.34±0.33 |
| | ✓ | ✓ | 70.31±0.45 | 85.22±0.29 | 74.22±0.50 | 87.55±0.34 |
| Swin-Tiny | | | 70.82±0.45 | 84.39±0.31 | 76.37±0.48 | 87.46±0.35 |
| | ✓ | | 71.18±0.46 | 86.00±0.29 | 76.71±0.48 | 88.88±0.32 |
| | | ✓ | 71.19±0.46 | 85.72±0.29 | 76.60±0.48 | 88.71±0.32 |
| | ✓ | ✓ | 71.31±0.45 | 86.07±0.29 | 77.35±0.48 | 89.49±0.31 |

compute the distance between two sets of features. We note that, compared with the baseline empirical MMD, ADGM is effective in improving the classification accuracy in all settings between the maximal and minimal improvements of 3.87% and 0.49%, indicating that our AMMD outperforms empirical MMD for few-shot classification. In addition, even using only ADGM to estimate the empirical distributions of query features or support features are both able to improve the classification accuracy within the maximal and minimal improvements of 3.78% and 0.07%. The results in Table 3 show that ADGM significantly improves the performance of the MMD-based few-shot classification metric, which may be due to the fact that AMMD focuses more on discriminative patch-level features, as shown in Fig. 3.

**Influence of the number of patches.** In using ResNet-12 to extract patch-level features of images, we need to first determine the grid strategy to split the images into patches, where the number of patches corresponds to the grid strategy. In order to explore the influences of patches with different parameter settings on the performance of few-shot classification by ResNet-12, we conduct experiments with AMMD under different grid strategies. As shown in Table 4, simply increasing the number of grid cells does not

Table 4: The 5-way few-shot classification accuracy (%) for the different patch configurations.

| Num. of patches | *mini*ImageNet | | *tiered*ImageNet | |
|---|---|---|---|---|
| | 1-shot | 5-shot | 1-shot | 5-shot |
| $4 \times 4$ | 68.77 | 84.02 | 72.60 | 85.13 |
| $3 \times 3$ | 69.35 | 84.68 | 73.40 | 86.73 |
| $2 \times 2$ | 68.82 | 85.02 | 71.93 | 86.83 |
| $4 \times 4 + 3 \times 3$ | 69.99 | 84.81 | 73.69 | 86.74 |
| $4 \times 4 + 2 \times 2$ | **70.65** | 85.16 | 74.09 | 86.85 |
| $3 \times 3 + 2 \times 2$ | 70.31 | **85.22** | **74.22** | **87.55** |

always improve the performance. We analyze that splitting the images into small grid cells will result in small local patch containing limited discriminative information for ResNet-12. Moreover, the multiscale-grid strategies achieve better results than the single-scale grid strategies, which may be due to the fact that multi-scale patches are more likely to cover discriminative local regions and contain richer multi-scale features. Considering the computational overhead and overall performance, we set $3 \times 3 + 2 \times 2$ as the number of patches for ResNet-12.

## 5 CONCLUSION AND LIMITATION

In this paper, we propose an AMMD distance for the few-shot classification. The proposed AMMD assists MMD with an attentive distribution generation module (ADGM). The ADGM adaptively estimates the distribution of patch-level features and is learned to put a larger probability mass on more discriminative patch-level features. Compared with the empirical MMD, AMMD emphasizes more on the discriminative features when measuring the distance between images in the feature space. Comprehensive experimental results demonstrate that the proposed AMMD is effective for few-shot image classification.

One limitation of the AMMD is that we assume that the distributions of image features are discrete distributions over image patches, which may be not aware of the unseen features in the continuous feature space. In the future, we will explore modeling the feature distributions by generative models that can model continuous distributions over finite image features.

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

# A APPENDIX

## A.1 DATASET DETAILS

For a thorough comparison with previous works, four widely adopted benchmark datasets are used in the paper. *mini*ImageNet consists of 100 classes with 600 images per class, which is divided into 64 classes as the meta-training set, 16 classes as the meta-validation set, and 20 classes as the meta-test set. *tiered*ImageNet contains 608 classes from 34 super-classes with a total of 779,165 images, which are split into 351 classes (20 super-classes) for meta-training, 97 classes (6 super-classes) for meta-validation, and 160 classes (8 super-classes) for meta-test. **CIFAR-FS** has 100 classes with each class containing 600 image images, which is divided into 64, 15, and 20 classes for meta-training, meta-validation, and meta-testing, respectively. **FC100** also has 100 classes with 600 images per class and we split it into 60, 20, and 20 classes.

## A.2 PRE-TRAINING

Similar to existing works (Wertheimer et al., 2021; Xie et al., 2022; Guo et al., 2022; Zhang et al., 2023), we pre-train ResNet-12 on the meta-training datasets, with the aim of recognizing images among all classes of the meta-training datasets (e.g., 64 classes on *mini*ImageNet). On the four benchmark datasets, we employ the SGD optimizer with an initial learning rate of 0.1. On *mini*ImageNet, we run 350 epochs with a batch size of 128, and the learning rate is decayed by multiplying 0.1 at 200 and 300 epochs. On CIFAR-FS, the pre-training process runs for 350 epochs with a batch size of 128, and we decay the learning rate for SGD with a multiplier 0.1 at 100, 200, 250, 300 epochs. On *tiered*ImageNet and FC100, we pre-train ResNet-12 for 120 epochs with a batch size of 128.

As done in FewTRUE (Hiller et al., 2022), we employ a self-supervised method called iBOT (Zhou et al., 2022) to pretrain Swin-Tiny. By formulating masked image modeling as knowledge distillation, iBOT learns to predict the features of masked patches of the images. To this end, we feed the original images into the teacher network and feed the masked images into the student network. The student network is taken as Swin-Tiny and the teacher network is the exponential moving average of the student network. Furthermore, the network of Swin-Tiny is trained by AdamW with a batch size of 512. We pre-train Swin-Tiny for 400 epochs on *tiered*ImageNet and for 800 epochs on the other three datasets including *mini*ImageNet, CIFAR-FS, and FC100. During the first 10 epochs, the learning rate is linearly increased to $5e^{-4} \times \frac{\text{batchsize}}{256}$. Subsequently, the learning rate is decreased to $1e^{-6}$ with a cosine schedule.

## A.3 META-TRAINING

For ResNet-12, as done in MCL (Liu et al., 2022) and DeepEMD (Zhang et al., 2023), we obtain patches by cropping each image into the grid of size $3 \times 3 + 2 \times 2$ and resize each patch as $84 \times 84$. By feeding patches into ResNet-12, we can extract $3 \times 3 + 2 \times 2$ patch-level features from each image. On all datasets, we run 40 epochs using SGD with initial learning rate 2e-4 (resp. 5e-4) and decay with a multiplier 0.5 every 10 epochs for 5-way 1-shot (resp. 5-way 5-shot) classification. Besides, we select the values of margin $\gamma$ and temperature $\tau$ by the performance of our AMMD in meta-validation and set $\gamma = 0.5$ and $\tau = 0.2$ (resp. $\tau = 0.5$) for 5-way 1-shot (resp. 5-way 5-shot) classification, except $\gamma = 0.3$ on *tiered*ImageNet.

For Swin-Tiny, as done in FewTRUE (Liu et al., 2021b), we resize each image as $224 \times 224$ and then feed it to the transformer architecture to obtain $7 \times 7$ patch-level features. On *mini*ImageNet, we run 40 epochs using SGD with initial learning rate of 5e-4 and decay it with a multiplier 0.5 every 10 epochs. On the other three datasets, we run 100 epochs using SGD with initial learning rate of 1e-3 and decay it with a multiplier 0.5 every 20 epochs. We set $\gamma = 0.3$ and $\tau = 0.2$ (resp. $\tau = 0.5$) for 5-way 1-shot (resp. 5-way 5-shot) classification, except $\tau = 0.02$ on *tiered*ImageNet for 5-way 1-shot classification. We further analyze the influence of temperature $\tau$ to the classification performance in Appendix A.4.

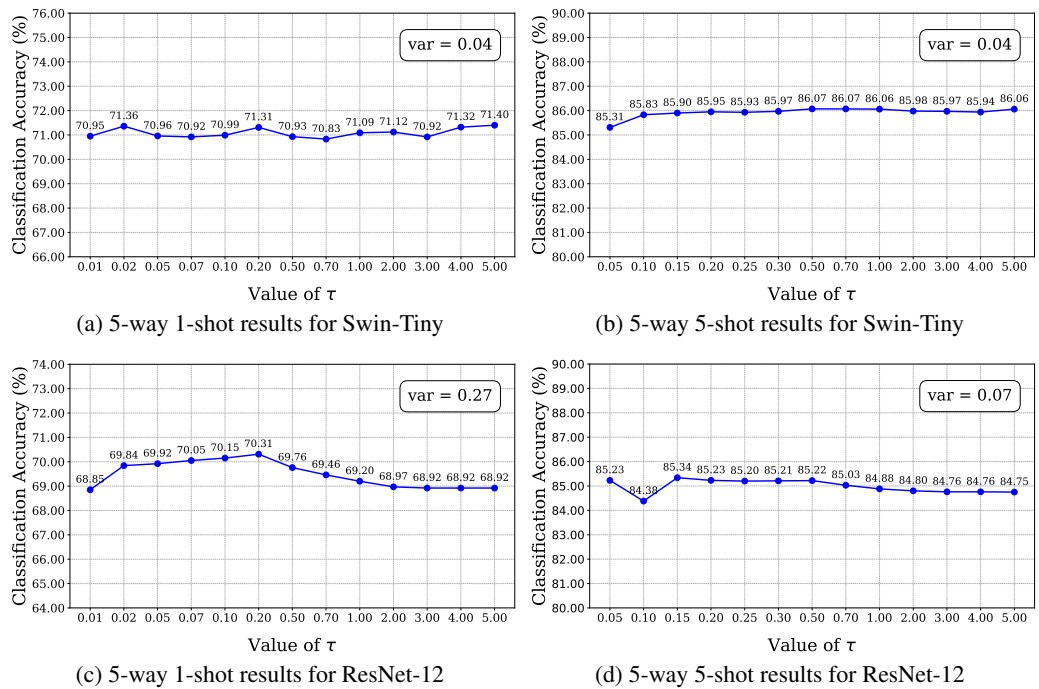

Figure A-1: Effection of the temperature $\tau$ for classification performance on *mini*ImageNet.

### A.4 ADDITIONAL EXPERIMENTS

**Influence of temperature $\tau$.** We analyze the influence of the temperature $\tau$ (in Eq. (9) and Eq. (10)) with different values for the proposed AMMD, by showing the curves of the few-shot image classification performance with different values of $\tau$. As shown in Fig. A-1, the classification accuracy of AMMD on *mini*ImageNet is generally stable, which indicates that our AMMD is not very sensitive to the temperature $\tau$.

Table A-1: Few-shot classification accuracy (%) of different kernel for AMMD.

| Backbone | Kernel | *mini*ImageNet | | *tiered*ImageNet | |
|---|---|---|---|---|---|
| | | 5-way 1-shot | 5-way 5-shot | 5-way 1-shot | 5-way 5-shot |
| ResNet-12 | Linear | 70.31±0.45 | 85.22±0.29 | 74.22±0.50 | 87.55±0.34 |
| | Multi-Gaussian | 68.73±0.48 | 83.27±0.32 | 71.86±0.52 | 87.03±0.34 |
| Swin-Tiny | Linear | 71.31±0.45 | 86.07±0.29 | 77.35±0.48 | 89.49±0.31 |
| | Multi-Gaussian | 70.28±0.43 | 85.22±0.30 | 76.63±0.48 | 88.34±0.32 |

**Influence of kernel type.** We employ the linear kernel for AMMD in this paper and we further explore the classification performance of employing AMMD with the multi-Gaussian kernel (Long et al., 2015). The results of using different kernel for AMMD as shown in Table A-1, the classification performance using the multi-Gaussian kernel is generally lower than using the linear kernel in the 5-way 1-shot setting and 5way-5-shot setting.

