# OpenReview forum: "AMMD: Attentive Maximum Mean Discrepancy for Few-Shot Image Classification"
_ICLR.cc/2024/Conference — ICLR 2024 Conference Withdrawn Submission_

### Official Review · Reviewer_wPps · 2023-10-28

**Soundness:** 2 fair
**Presentation:** 3 good
**Contribution:** 3 good
**Rating:** 5
**Confidence:** 5

**Summary:**

This paper deals with the few-show image classification problem. It proposes an attention-based module to adaptively weight the distributions for metric learning. The experiments show that the proposed method can generate feature distributions for computing maximum mean discrepancy, with higher probability mass on the discriminative local features.

**Strengths:**

1. The motivation is sound, which is weighting the local features to generate distributions for few-shot learning.

2. Visualization results show that the method assigns large weights for more discriminative local features.

3. The paper is well-written and easy to follow.

**Weaknesses:**

1. My biggest concern is that the performance of the proposed method is not SOTA based on numerical results. The most compared methods in this paper are published in AAAI2023 and PAMI, while the method in PAMI was first proposed in CVPR2020, which all seem a little bit outdated. The authors should consider comparing their methods with recent methods [1,2] published in CVPR2023 and ICCV2023. As for the numerical results, the accuracy under miniImageNet and tieredImageNet is 71.97±0.65, 87.06±0.38, 76.93±0.70, 90.12±0.45 while the number in this paper is 71.31±0.45 86.07±0.29 77.35±0.48 89.49±0.31. Furthermore, the authors use the Swin-Tiny backbone, which should achieve higher performance than ViT-S used by [2] as stated by the authors. Also, in paper [3], the reported accuracy under LR-DB-ventral in their Table 1 is 79.48 ± 0.26 and 91.03 ± 0.41 compared with 77.35±0.48 and 89.49±0.31 in this paper.

2. The contribution is not enough. As for me, the pipeline follows the same pipeline in the few-shot image classification field. The novelty lies in a re-weighting module for metric learning, which is based on attention. The techniques are not very novel and I think the contribution is not enough for the ICLR conference.

3. Experiments using WRN28 as the backbone should be carried as many papers including [3,4] in few-shot classification using this backbone.

4. (Minor) The citation of the paper "SpatialFormer: Semantic and Target Aware Attentions for Few-Shot Learning" is wrong. The authors are wrong.

5. (Minor) Authors should consider open-sourcing their codes.


[1] Transductive Few-shot Learning with Prototype-based Label Propagation by Iterative Graph Refinement.

[2] Class-Aware Patch Embedding Adaptation for Few-Shot Image Classification.

[3] Exploring Tuning Characteristics of Ventral Stream’s Neurons for Few-Shot Image Classifcation.

[4] Adaptive Distribution Calibration for Few-Shot Learning with Hierarchical Optimal Transport.

**Questions:**

1. What methods are using the reported accuracy in their papers and what methods are re-implemented by the authors in Table 1?

2. How can we select the most effective patch number in Table 4? Do the authors use validation sets to select this hyper-parameter?

3. As for me, the selected most discriminative parts are the objects and the unselected parts are the backgrounds. Can the authors explain why using the attentive module can only select the foreground objects?

---

### Official Review · Reviewer_qKBe · 2023-10-30

**Soundness:** 2 fair
**Presentation:** 3 good
**Contribution:** 1 poor
**Rating:** 3
**Confidence:** 5

**Summary:**

This paper introduces MMD distance as an alternative metric to tackle few-shot learning problems. The MMD distance is calculated over query and support features of all patches of images. To give different weights for different patches, an attention module is then applied to produce weights dynamically, leading to AMMD. The experiments show the effectiveness of the proposed method.

**Strengths:**

- Introducing the MMD distance to few-shot learning is interesting, and the attentive module makes sense for few-shot learning.
- The writing is clear and easy to understand.

**Weaknesses:**

- The motivation for using MMD as a metric is not stated in the paper. It is not clear why using MMD is better than other metrics like BDC [1] and EMD [2] distance, since MMD distance does not considers correspondence between support and query patches.
- As the method uses both supervised and self-supervised pretraining, the comparisons are unfair (self-supervised pretraining has proven to be extremely helpful for few-shot learning, as evidenced by [3]). In addition, cutting images into patches can lead to much better performance for every method [4], so in order to achieve a fair comparison, the authors should do this for every compared method at test time (for non-patch-based methods, cut into patches and average their features).
- The performance of pretrain-only baseline is missing.
- As shown in [3,5], a strong backbone can be much better than any pure meta-learning method. It is not clear from the paper whether the proposed method can scale to large datasets. The authors should apply their method to more pretrained models (like CLIP and DINO-v2) trained on large datasets (just like what is done in [5], first pretrain and then tune with meta-learning), and see if the performance improves more than other methods using the same pretrained models. Only by doing this, we can see some values of this paper.
- The classes of images in Figure 5 are all from the training set of miniImageNet. Thus it is not surprising for the visualization results. The authors should give visualization results from the unseen test set.


[1] Joint Distribution Matters: Deep Brownian Distance Covariance for Few-Shot Classification. CVPR 2022.

[2] DeepEMD: Few-Shot Image Classification with Differentiable Earth Mover’s Distance and Structured Classifiers. CVPR 2020.

[3] A Closer Look at Few-shot Classification Again. ICMl 2023.

[4] Rectifying the shortcut learning of background for few-shot learning. NeurIPS 2021.

[5] Pushing the Limits of Simple Pipelines for Few-Shot Learning: External Data and Fine-Tuning Make a Difference. CVPR 2022.

**Questions:**

See weaknesses above.

---

### Official Review · Reviewer_sp9L · 2023-10-31

**Soundness:** 2 fair
**Presentation:** 2 fair
**Contribution:** 2 fair
**Rating:** 5
**Confidence:** 5

**Summary:**

The paper introduces a novel approach called Attentive Maximum Mean Discrepancy (AMMD) for few-shot classification. In traditional empirical Maximum Mean Discrepancy (MMD) methods, the mean of patch-level features is used to approximate expectations.
Instead, the proposed AMMD adaptively estimates distributions of patch-level features by an attentive distribution generation module (ADGM) and then computes the expectations in MMD based on the estimated distributions.
In a word, AMMD integrates the attention-based module ADGM and the distance metric MMD.
The proposed AMMD obtains good performance in few-shot classification.

**Strengths:**

1. The writing is easy to follow.
2. Applying Maximum Mean Discrepancy (MMD) in few-shot learning is interesting.
3. The performance is good.

**Weaknesses:**

1. The novelty of ADGM is incremental. CTX and STANet also performed cross-attention between support and query.
2. Please further explain the meaning of eq.(7)
3. Please further explain the meaning of α and β.
4. According to eq.(9), α is the similarity map of query features with global support features. According to eq.(10), β is the similarity map of support features with global query features. Then, in Fig.1, What is the meaning of using MMD to measure the similarity between α and β?
5. Add more ablation studies of MMD, such as comparing to ADGM+cosine distance.
6. add more visualization comparisons with other attention-based methods, such as DeepEMD, CTX, and STANet.
7. Show the visualization comparison with the baseline without ADGM.
8. (1) The authors' information of STANet is incorrect. (2) The citation of MixFSL is incorrect. It should be Mixture-based feature space learning for few-shot image classification.

**Questions:**

see the weaknesses

---

### Official Review · Reviewer_H34j · 2023-11-01

**Soundness:** 2 fair
**Presentation:** 1 poor
**Contribution:** 2 fair
**Rating:** 5
**Confidence:** 4

**Summary:**

This paper introduces a novel approach called Attentive Maximum Mean Discrepancy (AMMD) aimed at enhancing the Maximum Mean Discrepancy (MMD) method. The proposal involves adaptively estimating distributions using an attentive distribution generation module to assist MMD. The focus is on applying AMMD to few-shot learning, treating it as an AMMD metric learning problem. To implement this, the authors use part-based feature representation to model the AMMD between images.

The method incorporates a meta-learning technique to train the attentive distribution generation module of AMMD. This module generates feature distributions for computing MMD between images, giving more weight to the more discriminative features. During the meta-test phase, each query image is labeled based on the support class that exhibits the minimum AMMD to the query image.

Experiments are done on typical few-shot image classification datasets.

**Strengths:**

- Illustrative figures 2 and 3 are clear.

- The proposed method shows performance improvement on few-shot image classification.

**Weaknesses:**

1. The novelty is limited. The patch-level feature extractor is not a new idea. As for the Attention-based Distribution Generation Module (ADGM), it estimates the improtance of patch-level feature. What is the difference if you directly use multi-head attention to obtain updated patch-level features? In this way, the patch-level features of query samples are transformed along with the patch-level features of support samples. Then, one can apply MMD to calculate distance. Why bother to design ADGM? The authors are also suggested to provide empirical evidence.

2. The proposed AMMD contains many parameters. Will that be costly to train? Any comparison on time consumption?

3. The benefits of using patch-based features can be explained in more length. Why is it better than learning with sample-level features?
The authors may add a comparison with a related baseline [1] about this question.

[1] Distribution Consistency Based Covariance Metric Networks for Few-Shot Learning, AAAI-19


4. The paper needs significant proofreading.

- Some sentences are hard to follow, such as "each query image is labeled as the support class with minimal AMMD to the query image" in the abstract.

- Wrong method names, such as FewTrue and FewTURE. Even for the proposed method, the authors use a wrong name. E.g., AWGM in Table 3.

**Questions:**

Please see the weakness part.